# On the Effect of Cobalt Promotion over Ni/CeO$_2$ Catalyst for CO$_2$ Thermal and Plasma Assisted Methanation

**Golshid Hasrack** [1,2]**, Maria Carmen Bacariza** [2] **, Carlos Henriques** [2] **and Patrick Da Costa** [1,*]

[1] Institute Jean Le Rond d'Alembert, Sorbonne Université, CNRS UMR 7190, 2 Place de la Gare de Ceinture, Saint-Cyr-l'Ecole, 78210 Paris, France; golshid.hasrack@sorbonne-universite.fr

[2] Centro de Química Estrutural, Chemical Engineering Department, Instituto Superior Técnico, Universidade de Lisboa, Av. Rovisco Pais, 1049-001 Lisboa, Portugal; maria.rey@tecnico.ulisboa.pt (M.C.B.); carlos.henriques@tecnico.ulisboa.pt (C.H.)

[*] Correspondence: patrick.da_costa@sorbonne-universite.fr; Tel.: +33-1-4427-9562

**Abstract:** In recent years, carbon dioxide hydrogenation leading to synthetic fuels and value-added molecules has been proposed as a promising technology for stabilizing anthropogenic greenhouse gas emissions. Methanation or Sabatier are possible reactions to valorize the CO$_2$. In the present work, thermal CO$_2$ methanation and non-thermal plasma (NTP)-assisted CO$_2$ methanation was performed over 15Ni/CeO$_2$ promoted with 1 and 5 wt% of cobalt. The promotion effect of cobalt is proven both for plasma and thermal reaction and can mostly be linked with the basic properties of the materials.

**Keywords:** CO$_2$ methanation; Ni-based catalysts; plasma catalysis; Co promotion





## 1. Introduction

Greenhouse gases such as CO$_2$ are responsible for climate change and global warming. Thus, nowadays their control and utilization are an important subject from a scientific and economic point of view. CO$_2$ emissions produced from an energy system based on fossil fuels are the main reason for climate change [1]. To overcome this problem and decrease these emissions in the atmosphere, several alternatives have been analyzed. Carbon capture and utilization (CCU) or storage (CCS) are among the solutions which took attention in the last decade for this aim. These processes will help to either turn CO$_2$ emissions into valuable chemicals or make a closed cycle of carbon in industries (e.g., cement) where synthetic natural gas produced from their CO$_2$ emissions through Sabatier reaction could be re-used for combustion processes [2]. Among the CO$_2$ valorization strategies, CO$_2$-to-methanol [3,4], Reverse Water Gas Shift (RWGS) reaction [5] and CO$_2$ methanation [6] are the most studied. Indeed, methanolation and methanation are mature solutions already industrially implemented [7,8].

Regarding CO$_2$ methanation, different types of metal-supported catalysts have been reported, with Ni, Ru, Co, Fe, Rh or Pd as the main active metals [9–14]. Among all, Ni-based catalysts are the most used due to their high activity, their availability, and their lower cost [15]. However, their catalytic activity depends on different factors, such as Ni loading, the type of support or the preparation method [16]. In terms of supports nature, alumina, silica, double layered mixed oxides, zeolites, ceria, or ceria-zirconia have been studied [6,11,14,17,18]. Among them, ceria (CeO$_2$) can be considered as a promising support because of its properties such as oxygen mobility, which enhances CO$_2$ activation and hinders carbon deposition [19,20].

Furthermore, the use of promoters such as Y, Mn, La, Cu or Co was reported as favorable for improving catalysts' activity and stability [17,21–24]. Among them, cobalt incorporation has not been widely explored so far. Indeed, on bimetallic Co-Ni/Al$_2$O$_3$ catalysts, Liu et al. [25] showed that catalytic properties were enhanced for CO$_2$ methanation by the formation of Co-Ni alloy, leading to a higher surface area and better Ni$^0$ dispersion.

Moreover, Alrafei et al. [16] also showed that for Co-Ni/$Al_2O_3$ catalytic systems, that the presence of Co improved $Ni^0$ dispersion and Ni species reducibility. Furthermore, Summa et al. [17] concluded that the addition of cobalt in low amounts (0.5–1 wt%) resulted in an optimum improvement of the surface properties, such as basicity and hydrogen uptake, which increased the catalytic performance for $CO_2$ methanation.

Besides the improvement of $CO_2$ methanation catalysts formulation, plasma assisted catalysis has been developed for more than 10 years for improving this process [24,26,27]. Among the types of plasma reported, the dielectric-barrier discharge (DBD) is the most commonly used for $CO_2$ methanation [24–28]. Moreover, as for thermal methanation, Ni based catalysts are the most used for methanation under plasma-assisted catalysis conditions [24–28]. In terms of used catalysts, different supports such as titania [29], ceria [19], zirconia [30], ceria-zirconia [31–34], alumina [35,36], zeolites [6,37,38], metal–organic frameworks [35] or mixed-oxides derived from hydrotalcites [18] have been reported. In terms of promoters, only cerium [6–18] and lanthanum [37] appeared to promote Ni-based catalysts performances for the DBD plasma methanation process over different supports. Accordingly, Chen et al. [37] demonstrated, compared to a non-promoted catalyst, that the addition of La resulted in an improvement of the turnover frequency and selectivity towards $CH_4$. More recently, it was shown on DBD plasma using ceria-zirconia-based Ni catalysts that the promotion of elements such Cu, Co, Mn, La, Y, Gd and Sr can considerably alter both the physicochemical and the electrical features of the catalysts, resulting in different plasma-catalytic performance. Indeed, authors reported a real improvement with the studied elements except for Cu and Sr [39].

Furthermore, to our knowledge, so far, no studies have dealt with Co-promoted Ni catalysts supported on $CeO_2$ for conventional thermal neither for plasma-assisted $CO_2$ methanation. Thus, in this study, the activity and stability of Co-Ni/$CeO_2$ using 2 Co loadings were investigated for $CO_2$ methanation and a correlation between structure and activity was proposed.

## 2. Results and Discussion

### 2.1. Characterization Results

TPR-$H_2$ profiles are presented in Figure 1. It is worth mentioning that 2 peaks were observed at 330–350 °C and 465–510 °C for Ni-containing catalysts, which could be attributed, based on literature, to the support reduction, more precisely to surface oxygen species in $CeO_2$ and $CeO_2$ bulk reduction, respectively [19].

One can also note that, with the introduction of both Ni and Co, $CeO_2$, peaks shift towards lower temperatures. Moreover, for all the studied Ni catalysts, a well-defined peak is found at 200–300 °C, which shifts towards higher temperatures with increasing Co loadings. This peak is generally attributed to easily reducible NiO species [19]. Finally, a significant reduction peak is observed at ~500 °C on 5Co15Ni/$CeO_2$ catalyst, which could be attributed to the reduction of $CoO_x$ or Co cationic species [40]. As reported in Table 1, the peaks of reduction identified in this study are comparable to those found in previous works. In addition, total $H_2$ consumptions for the catalysts from this work are also reported in Table 2, with increasing Co loadings leading to higher values.

**Table 1.** Comparison of reduction peaks for similar catalysts with different contents with present study.

| Catalyst | First Main Peak Temperature (°C) | Second Main Peak Temperature (°C) | $H_2$ Consumption [mmol/g] | References |
|---|---|---|---|---|
| $CeO_2$ | 443 | - | 504 | Present Study |
| 15Ni/$CeO_2$ | 268 | 401 | 1086 | Present Study |
| 7.5Ni/$CeO_2$ | 282 | 400 | - | [19] |
| 1Co15Ni/$CeO_2$ | 274 | 401 | 1267 | Present Study |
| 5Co15Ni/$CeO_2$ | 300 | 426 | 1575 | Present Study |
| 3.75Co3.75Ni/$CeO_2$ | 295 | 420 | - | [19] |

$CO_2$-TPD profiles of the studied samples are shown in Figure 2. The profile of the reference sample ($15Ni/CeO_2$) presents three main peaks at 138 °C, 206 °C and 400 °C, which can be ascribed to weak, medium, and strong basic sites, respectively. However, in the presence of Co only two peaks can be observed. The total basicity and the weak, medium, and strong basicity repartition are reported in Table 2.

**Table 2.** Comparison of basicity of the samples in present study.

| Catalyst | Weak Basic Sites [µmol/g] | Medium Basic Sites [µmol/g] | Strong Basic Sites [µmol/g] | Total Basic Sites [µmol/g] |
|---|---|---|---|---|
| $CeO_2$ | 88 | 16 | - | 103 |
| $15Ni/CeO_2$ | 33 | 112 | 159 | 304 |
| $1Co15Ni/CeO_2$ | 67 | 212 | - | 279 |
| $5Co15Ni/CeO_2$ | 107 | 80 | - | 187 |

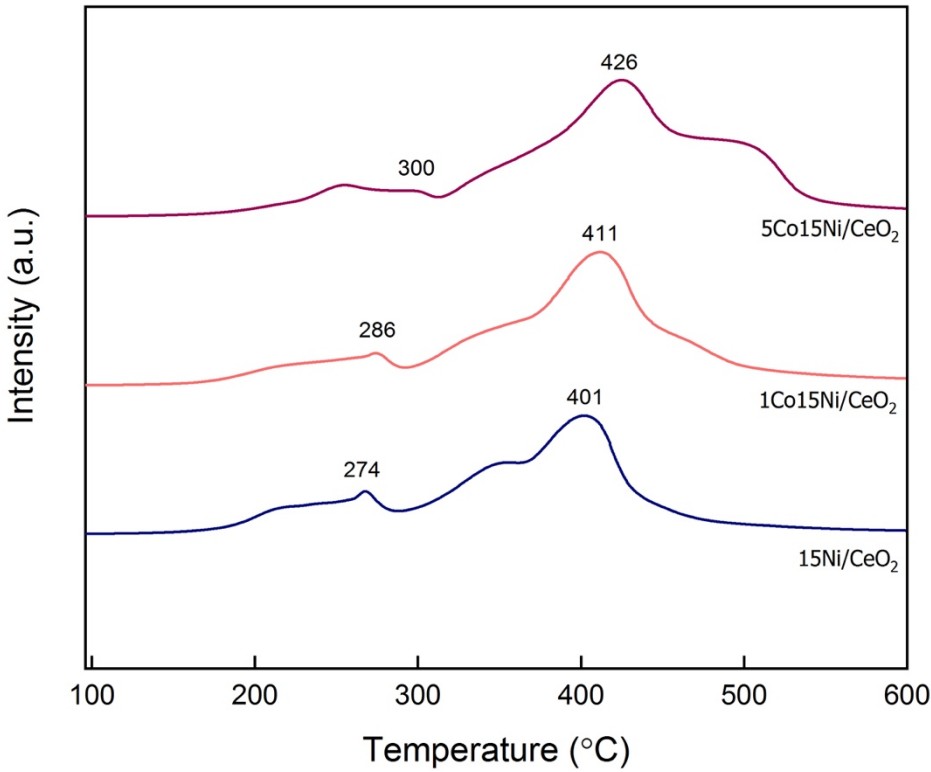

**Figure 1.** $H_2$-TPR profiles for calcined samples with different Co content.

It is worth mentioning that Co addition led to a decrease in the total basicity, while the number of medium-strength basic sites increased in presence of 1 wt% Co and later decreased with 5 wt% Co. This might indicate that the addition of excess Co may cover some basic sites, resulting in an adverse effect on the $CO_2$ chemisorption [25]. Additionally, it is observed that desorption peak temperatures shift towards higher values with the increase in Co content, suggesting a stronger interaction between $CO_2$ and the active sites [17]. When compared with the literature, a Ni-Co catalyst supported over an hydrotalcite (HT1Co20Ni) [17] presented lower medium-strength basic sites compared to the $1Co15Ni/CeO_2$ catalyst of this study. As the medium basic sites are the most relevant for $CO_2$ methanation [41], it can be concluded that, among the studied catalysts, $1Co15Ni/CeO_2$ presents the best basic properties.

In terms of textural properties (Table 3), assessed by $N_2$ adsorption, the incorporation of metals over ceria support led to a reduction of the total pore volume ($V_P$) and the BET surface area ($S_{BET}$). When comparing the samples with 1 and 5 wt% Co, no significant differences in terms of textural properties were found, suggesting that the variation of Co loading did not induce a remarkable effect on these parameters.

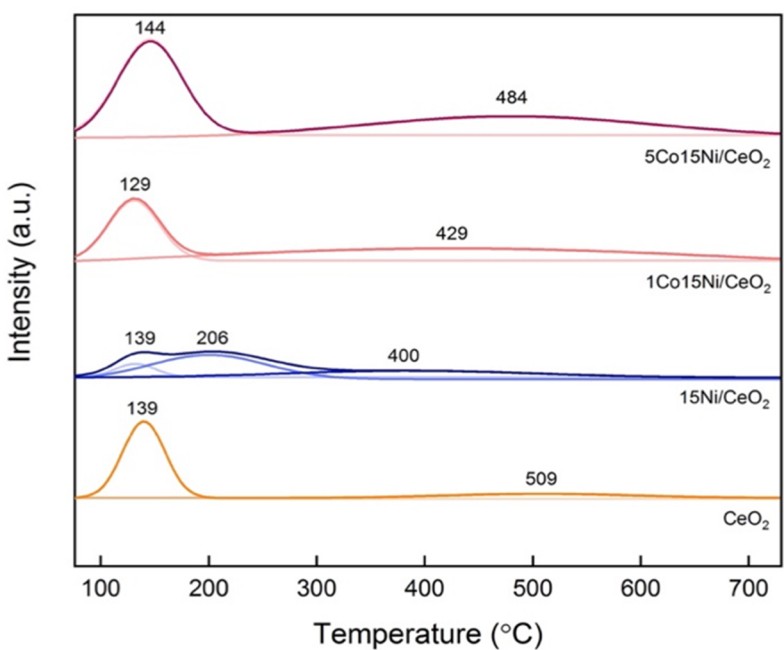

**Figure 2.** $CO_2$-TPD profiles for reduced catalysts with different Co content.

**Table 3.** Porous volume and external surface area for studied catalysts.

| Catalyst | $V_P$ (cm$^3$ g$^{-1}$) | $S_{BET}$ (m$^2$ g$^{-1}$) |
|---|---|---|
| CeO$_2$ | 0.19 | 258 |
| Ni/CeO$_2$ Reduced | 0.14 | 127 |
| Ni-1Co/CeO$_2$ Reduced | 0.13 | 108 |
| Ni-5Co/CeO$_2$ Reduced | 0.13 | 107 |

In addition, XRD diffractograms of calcined and reduced catalysts are presented in Figure 3. The presence of CeO$_2$ diffraction lines (28.6, 33.1, 47.5, 56.4, 59.1, 68.1, 76.8 and 79.1°; JCPDS card 81-0792 [19,42]) is confirmed in all of the catalysts after calcination and reduction, indicating that Ce species nature was not affected by the reduction process. Regarding Ni species, NiO diffraction lines were clearly identified in calcined Ni and Co-Ni catalysts at 37, 43.2 and 62.9° (JCPDS card 78-0643) [19], while Ni$^0$ phases were present after reduction (44.7 and 51.8°) [19]. We note that no Co$_3$O$_4$ diffraction lines (JCPDS card 78-1970) were identified in Co-containing catalysts. This could be due to the low metal loading used (1–5 wt%) or the presence of highly dispersed Co oxy-species on the catalysts. Based on CeO$_2$ diffraction lines, an average crystallite size of 5 nm was determined (Table 4). Regarding NiO crystallite sizes, results are shown in Table 5 and suggest that Co incorporation leads to the formation of larger NiO crystallites after calcination (increase of 5 nm when comparing to 15Ni/CeO$_2$). Furthermore, for reduced catalysts, the incorporation of 5 wt% Co led to the formation of smaller Ni$^0$ crystallites (32 nm, lower than the 38–39 nm obtained for 15Ni/CeO$_2$ and 1Co15Ni/CeO$_2$ reduced catalysts; Table 4), indicating that Co could have an efficient effect in the prevention of agglomeration processes in Ni$^0$ particles. Furthermore, when compared to other studies, it is worth mentioning that Benrabbah et al. [43] reported values of ~28 nm after reduction for 15Ni/CeZrO$_2$. The higher values obtained in this work could be due to the preparation method, as these authors used a heating rate for calcination of 5 °C/min while in this study this step was carried out by 10 °C/min.

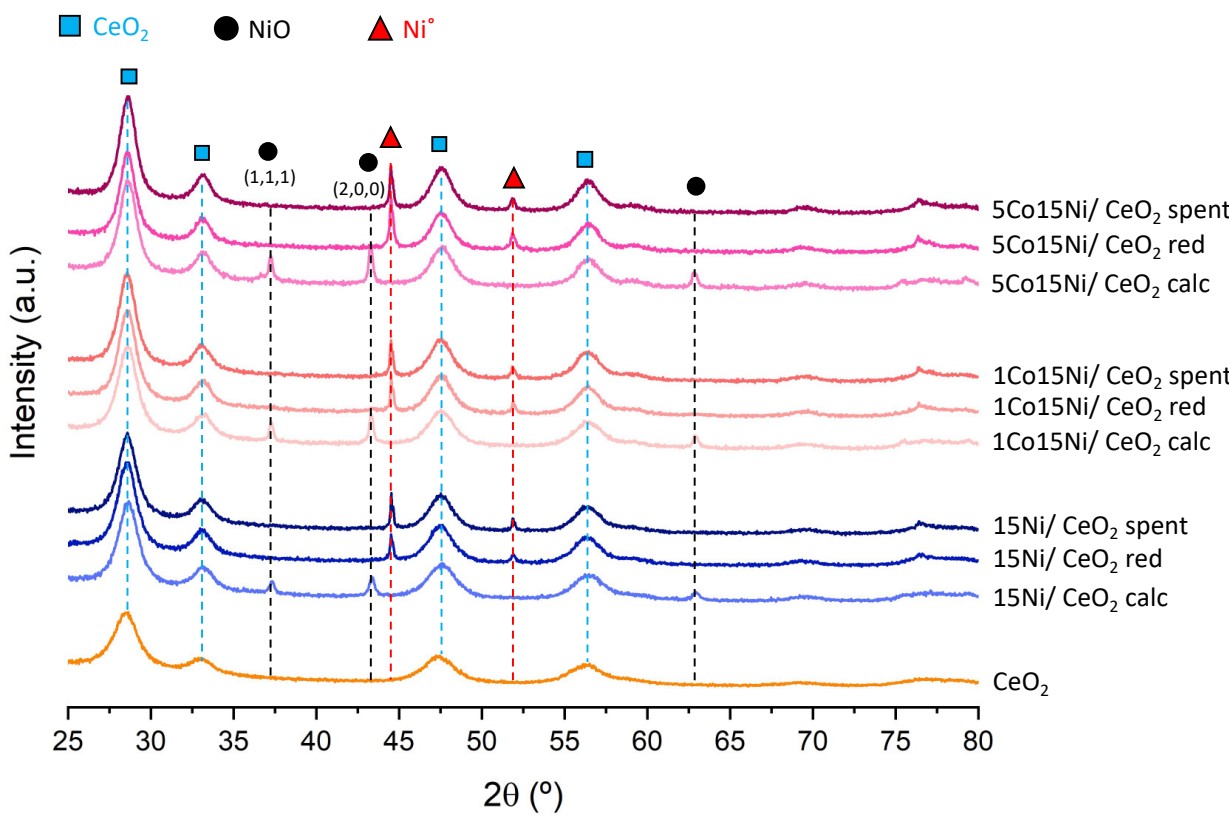

**Figure 3.** X-ray diffractograms of the studied catalysts.

**Table 4.** NiO and Ni$^0$ crystallite sizes after calcination, reduction and reaction determined applying Scherrer equation.

| Catalyst | d$_{NiO}$ (nm)$_{Calcined}$ | d$_{Ni}^0$ (nm)$_{Reduced}$ | d$_{Ni}^0$ (nm)$_{spent}$ |
|---|---|---|---|
| 15Ni/CeO$_2$ | 26 | 38 | 45 |
| 1Co15Ni/CeO$_2$ | 31 | 39 | 38 |
| 5Co15Ni/CeO$_2$ | 31 | 32 | 34 |

**Table 5.** Comparison between Ni-Co catalysts with different supports (conversion and selectivity are all considered at 300 °C).

| Catalyst | Conversion (%) | Selectivity (%) | GHSV (h$^{-1}$) [b] | Ni$^0$ Size (nm) | Basicity (W/M/S) [c] (µmol/g) | | | Ref. |
|---|---|---|---|---|---|---|---|---|
| 5Co15Ni/Al$_2$O$_3$ | 74 | - | 9554 | - | - | - | - | [16] |
| 5Co20Ni/Al$_2$O$_3$ | 85 | - | 9554 | 13 | - | - | - | [16] |
| 10Co10Ni/Al$_2$O$_3$ | 74 | - | 9554 | 31 | - | - | - | [16] |
| 3Co10Ni/Al$_2$O$_3$ [a] | 15 | 100 | 10,000 | 3 | - | - | - | [25] |
| 2Co8Ni/Al$_2$O$_3$ [a] | 60 | 93 | 15,000 | - | - | - | - | [44] |
| HT1Co20Ni | 77 | 99 | 12,000 | 16 | 18 | 66 | 94 | [17] |
| 1Co15Ni/CeO$_2$ | 82 | 98 | 52,000 | 39 | 67 | 212 | 0 | Present study |
| 5Co15Ni/CeO$_2$ | 79 | 95 | 52,000 | 32 | 107 | 80 | 0 | Present study |

[a] Ordered mesoporous alumina; [b] Gas hourly space velocity; [c] Weak/Medium/Strong basic sites.

TEM was then used to analyze the $Ni^0$ dispersion on the support. Figure 4 shows TEM micrographs of the catalysts from this work with different resolutions. Due to the low contrast of the support, it was difficult to conclude on the distribution of $Ni^0$ particles, so that EDS-TEM technique was also carried out (Figure 5). Based on the obtained results, it could be concluded that cobalt incorporation improves the dispersion of both Ni and Co species. However, the calculation the particle size distribution was not possible due to the low contrast between the metals and the support.

## 2.2. Catalytic Results

Figure 6a reports the $CO_2$ conversion as a function of temperature for the studied catalysts after a pre-reduction at 500 °C. One can note that, at low temperature (250 °C), the $15Ni/CeO_2$ catalyst shows the lower activity with a conversion of $CO_2$ of 16%. However, $1Co15Ni/CeO_2$ and $5Co15Ni/CeO_2$ catalysts presents a significantly high conversion of around 74% and 62%, respectively, at the same temperature. By increasing the temperature, the conversion for $15Ni/CeO_2$ reaches 80%. On the other hand, the promoted catalysts show higher conversions, up to 85%.

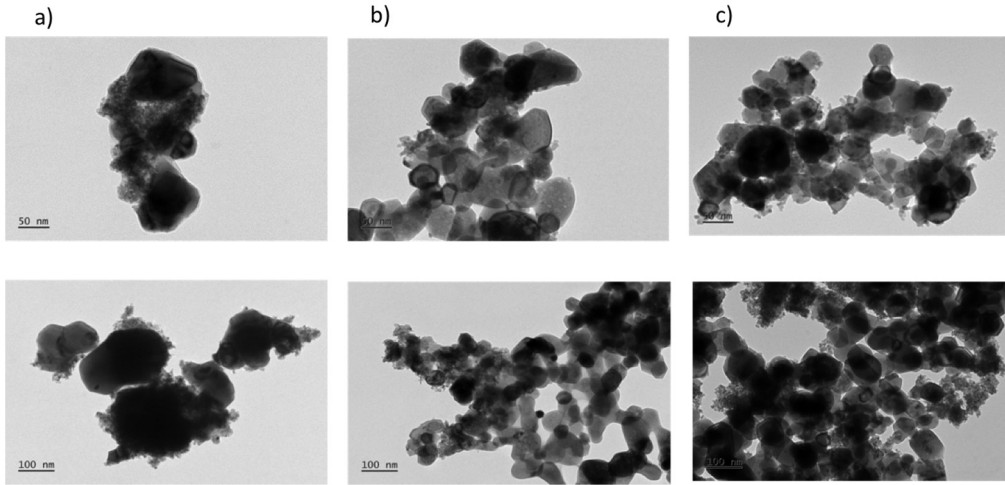

**Figure 4.** TEM images of (**a**) $15Ni/CeO_2$; (**b**) $1Co15Ni/CeO_2$; (**c**) $5Co15Ni/CeO_2$ with different scales after reduction.

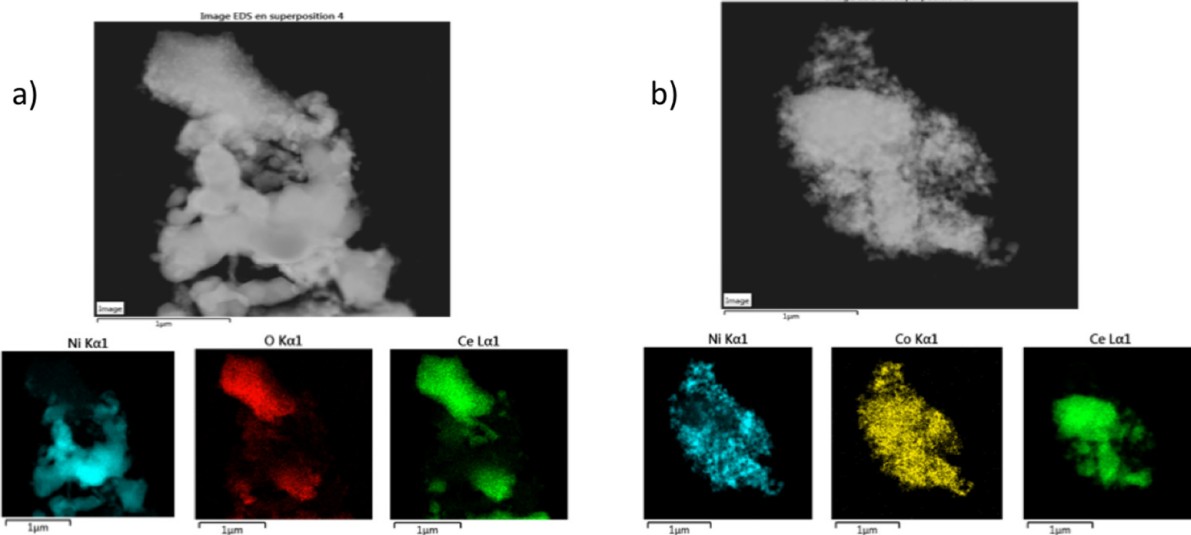

**Figure 5.** EDS-TEM images of (**a**) $Ni/CeO_2$; (**b**) $Co-Ni/CeO_2$ after reduction.

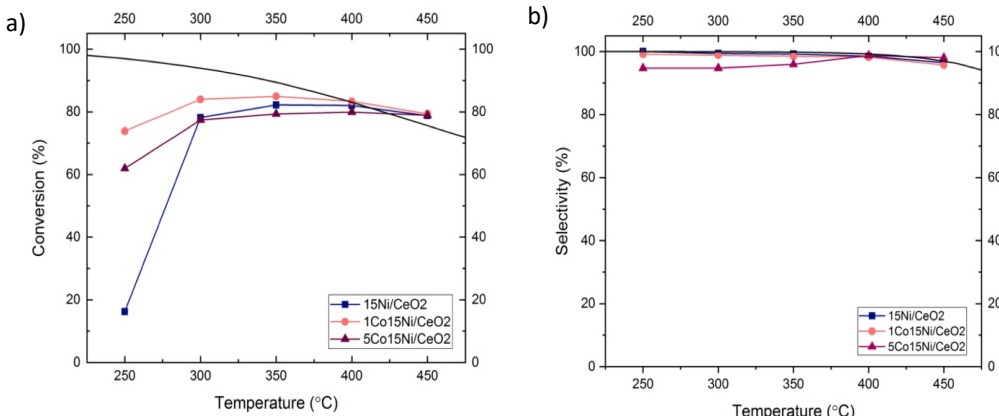

**Figure 6.** (**a**) Conversion and (**b**) selectivity of Co-Ni/CeO$_2$ catalyst in thermal CO$_2$ methanation with different Co loadings.

Moreover, it is worth noting that the highest conversion was exhibited at 350 °C by 1Co15Ni/CeO$_2$. Figure 6b reports CH$_4$ selectivity of all of the samples. For 15Ni/CeO$_2$ and 1Co15Ni/CeO$_2$ catalysts, the selectivity is close to 100% while for 5Co15Ni/CeO$_2$ the selectivity is around 94% at 250 °C and then with the increase in the temperature (350 °C) it reaches 100%. According to the literature, at a high temperature, reverse water gas shift starts to simultaneously occur with methanation. Then, methanation of CO begins, resulting in a decrease in CH$_4$ selectivity [5]. Additionally, it could be expected that the catalyst with the higher H$_2$ consumption has higher activity as H$_2$ chemisorption plays an important role in CO$_2$ methanation [25]. However, CO$_2$ conversion is much higher for 1Co15Ni/CeO$_2$ catalyst. Furthermore, there is another factor which has a remarkable impact. According to the result of number of basic sites, with the increase in Co loading the number of basic sites decreases, suggesting the covering of some sites by Co species, as previously discussed [25]. On the other hand, the promotion of 15Ni/CeO$_2$ catalyst with 1% Co enhanced the number of medium strength sites, which present a more beneficial effect in the reaction [41], which could explain the more favorable performances exhibited by 1Co15Ni/CeO$_2$ catalyst.

After the tests, catalysts were analyzed by XRD (Figure 3), being only Ni$^0$ diffraction lines and no NiO phases observed. This indicates that no reoxidation of Ni$^0$ species occurred during the experiments. Regarding Ni$^0$ crystallite sizes before and after the reaction (Table 4), the differences (1–2 nm) are negligible for 1 and 5 wt% Co-containing catalysts, which means that no remarkable sintering processes occurred. On the contrary, for 15Ni/CeO$_2$ catalyst, an increase in the Ni$^0$ crystallite size from 38 nm to 45 nm before and after reaction, respectively, was observed. This indicates that Co presents a positive effect in the prevention of Ni$^0$ sintering processes. When comparing to literature for monometallic Ni catalysts, Mikhail et al. [45] studied 15Ni/CeZrO$_2$ catalysts and found out that Ni$^0$ size increased from ~24 to ~46 nm after reaction, which was attributed to the occurrence of sintering. In other studies, regarding Ni-based zeolite catalysts from Guo et al. [46], an increase in Ni$^0$ particle size from ~14 to ~29 nm was observed for 10Ni/ZSM-5 before and after reaction, being this again attributed to sintering effects. Consequently, the increase in the Ni$^0$ crystallite size verified for 15Ni/CeO$_2$ catalyst is in accordance with the results in the literature.

Spent samples were also analyzed by TGA (Figure 7). As observed, a mass variation process was observed below 400 °C in the catalysts, corresponding to the loss of adsorbed water in the samples. On the other hand, at higher temperatures (>600 °C), where coke should be decomposed, the mass loss is negligible, indicating that no carbonaceous species were deposited in the catalysts during the tests [47].

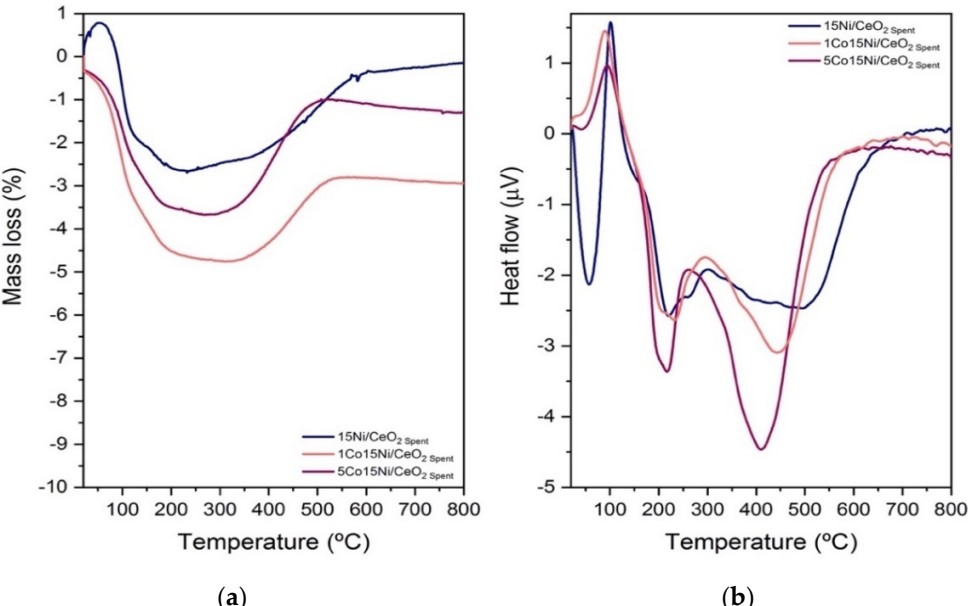

(**a**)                                                      (**b**)

**Figure 7.** TGA analysis with $O_2$ for (**a**) Mass loss and (**b**) Heat loss of Co-Ni/CeO$_2$ catalyst with different Co content.

Furthermore, stability tests were performed for the studied samples, being the results presented in Figure 8. Tests were carried out at 250 °C for 5 h with the same conditions used in the tests performed at variable temperatures. It can be observed that in the first 20 min all samples lose some activity (<9%). However, afterwards, all of the catalysts became stable. Moreover, the stability tests showed that 1Co15Ni/CeO$_2$ catalyst presented the higher activity.

Table 5 presents the results obtained for Co-Ni samples with different supports under CO$_2$ methanation conditions at 300 °C for different catalysts. When comparing 5Co15Ni/CeO$_2$ catalyst to 5Co15Ni/Al$_2$O$_3$ [16], which has the same loadings for the active metal and promoter, it can be seen that the conversion is higher by 5%, while GHSV for the catalyst of this study is much higher than that used for the alumina support. This indicates that 5Co15Ni/CeO$_2$ catalyst presents a higher efficiency.

Moreover, for the 1Co15Ni/CeO$_2$ catalyst, when comparing to HT1Co20Ni [17], higher conversion with much higher GHSV can be observed. It is worth mentioning that when comparing the basicity of these two catalysts, the number of basic sites (especially medium ones) for the 1 wt% Co catalyst of this study is higher, which is in a good agreement with the reported effect of the catalyst's basicity on the performances [41].

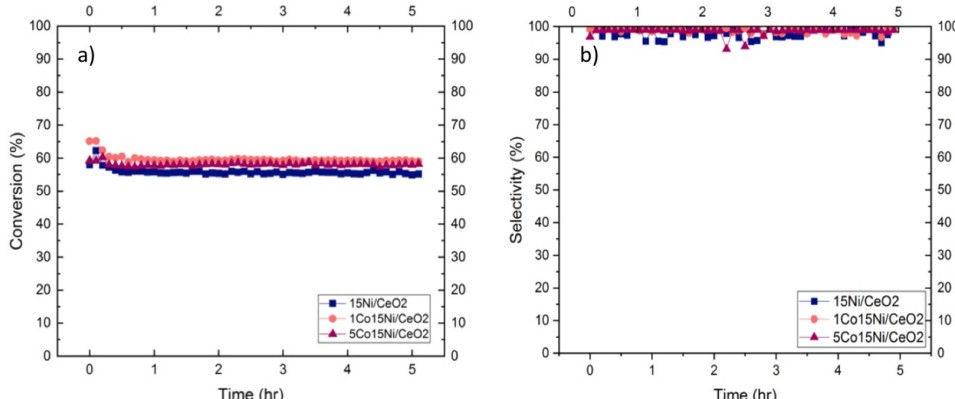

**Figure 8.** Stability tests: (**a**) CO$_2$ Conversion and (**b**) CH$_4$ Selectivity of Co-Ni/CeO$_2$ catalysts for 5 h run at 250 °C.

### 2.3. Plasma Assisted Catalytic Methanation

Figure 9a–c report the $CO_2$ conversion as a function of voltage applied in keeping the frequency constant and equal to 12.3 kHz. We note that for all the studied catalysts that the conversion of $CO_2$ and the methane selectivity increased with the increase in the voltage; similarly to the thermal catalysis, as reported in the Figure 9a. the $15Ni/CeO_2$ catalyst showed the lower activity with a conversion of $CO_2$ of 16% at low voltage. However, for higher voltage the conversion of $CO_2$ for $15Ni/CeO_2$ catalyst is higher than the one obtained for $5Co15Ni/CeO_2$ one.

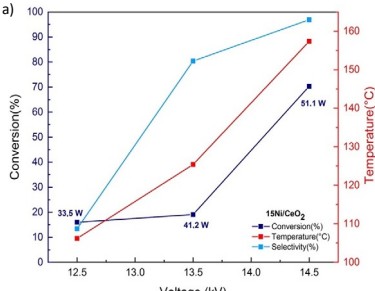 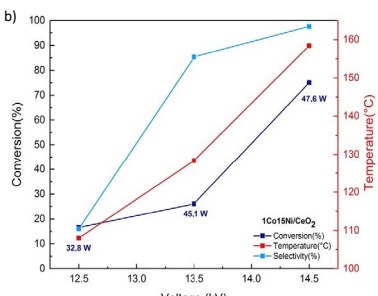 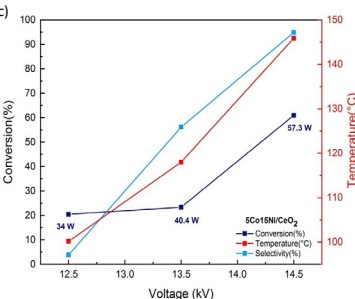

**Figure 9.** $CO_2$ conversion, Temperature measured and $CH_4$ selectivity of (**a**) $15Ni/CeO_2$, (**b**) $1Co15Ni/CeO_2$, (**c**) $5Co15Ni/CeO_2$, catalysts as function of voltage.

Furthermore, for all the range of studied voltage from 25 to 29 kV peak to peak which corresponds to 12.5 kV to 14.5 kV peak voltage in Figure 9, the $1Co15Ni/CeO_2$ catalyst exhibited higher conversion (Figure 9b). This conversion is of course increase with the voltage. Finally, the higher conversion was found to be 75% at 29 kV. The conversion is also linked with the temperature measured for the different operating conditions. Indeed, the higher temperature was observed for the higher obtained conversion. It was also already reported on promoted Ni/CeZrOx catalysts [39] that depending on the type of the promoter, a modification in the basicity distribution could be observed. Thus, the investigation of basic properties showed that the addition of the promoters (Gd, Y and Co) not only led to an increase in the total number of basic sites of the catalyst, but also the number of moderate basic sites which are reported to be the predominant ones for methanation reaction. In our study, among the Co promoted materials the $1Co15Ni/CeO_2$ catalyst exhibited both the higher number of basic sites but also the higher number of medium basic ones. Thus, in agreement with thermal methanation, the basic sites presented on our materials play an important role both in thermal and plasma assisted methanation reaction on Ni-Co/CeO2 catalysts. On the other hand, the synergistic effects are also an important factor regarding different behavior of catalyst. Furthermore, the plasma can lower the activation barrier which leads to a faster reaction occurrence. Finally, the plasma can also change the surface reaction pathway with the presence of excited species which results in a change in the transition state, as proposed elsewhere [24].

## 3. Materials and Methods

### 3.1. Catalysts Synthesis

$Ni/CeO_2$ and $Co-Ni/CeO_2$ catalysts with different cobalt loadings were prepared by wet impregnation method. For this purpose, a certain amount of Ni nitrate hexahydrate ($Ni(NO_3)_2·6H_2O$) was dissolved in distilled water. Following this, a certain amount of commercial cerium oxide ($CeO_2$) and Co nitrate hexahydrate ($Co(NO_3)_2·6H_2O$) added to this solution. Then, the mentioned solution was stirred at 50 °C for 4 h and then dried at 100 °C overnight. Subsequently, the samples were calcined at 500 °C for 4 h in static air. Table 6 reports the studied catalysts and summarize the nominal metal loadings.

**Table 6.** Studied catalysts.

| Catalyst | Metals Loading (% wt) (Co/Ni/CeO$_2$) |
|---|---|
| 15Ni/CeO$_2$ | 0/15/85 |
| 5Co15Ni/CeO$_2$ | 5/15/80 |
| 1Co15Ni/CeO$_2$ | 1/15/84 |

### 3.2. Characterization Methods

Catalysts were characterized by temperature-programmed reduction with hydrogen (TPR-H$_2$), temperature-programmed desorption of CO$_2$ (CO$_2$-TPD), N$_2$ adsorption, X-ray diffraction (XRD), transmission electron microscopy (TEM) and thermogravimetric analysis (TGA).

TPR-H$_2$ experiments were carried out on a BELCAT-M equipped with a thermal conductivity detector (TCD) to characterize catalysts' reducibility (BEL Japan, Inc., Osaka, Japan). For this purpose, 60 mg of calcined sample were first degassed in helium atmosphere at 100 °C for 130 min and then reduced in 5% H$_2$/Ar mixture with a heating rate of 10 °C/min starting from 100 °C to 700 °C. TPD-CO$_2$ was performed after TPR-H$_2$ run, using the same device. CO$_2$ was adsorbed at 80 °C for 1 h from a mixture of 10% CO$_2$/He. Then, helium flow was applied for 15 min in order to desorb weakly adsorbed CO$_2$. Finally, the materials were heated from 80 °C to 800 °C in helium to analyze their basic properties based on the desorption temperature. TPD profiles were deconvoluted into three Gaussian peaks corresponding to weak, medium, and strong basic sites in agreement with literature [25].

N$_2$ adsorption was carried out for reduced catalysts on an Autosorb iQ equipment (Quantachrome, Odelzhausen, Germany) at −196 °C. Catalysts (80–100 mg) were degassed under vacuum prior to the experiments at 90 °C (1 h) and 350 °C (4 h). Total pore volumes (V$_P$) were measured at a relative pressure (p/p$_0$) of 0.97 while surface areas (S$_{BET}$) were obtained by BET method.

XRD patterns were obtained from a Bruker AXS Advance D8 diffractometer combined with a 1D detector (SSD 160), using a Ni filter (Bruxer, Oeiras, Portugal). The scanning range was from 5 to 80° (2θ). The step time and step size were 0.5 s and 0.03°, respectively. CeO$_2$, NiO and Ni$^0$ average crystallite sizes were calculated in applying the Scherrer equation [47]. For this purpose, the considered diffraction lines were: 28.6, 33.1, 47.5 and 56.4° for CeO$_2$; 37.3° and 43.3° for NiO; and 44.5 and 52.1° for Ni$^0$.

High-Resolution TEM was performed using an JEOL JEM-2010 equipment with EDS to characterize the deposited carbon and to obtain a precise analysis of the planes registered at the nanometer scale (JEOL, Gmbh, Freising, Germany). The specimens were prepared by dropwise addition of a colloidal solution in ethanol onto a copper grid covered with amorphous carbon film [47].

Finally, TGA analysis were carried out using a Setsys Evolution TGA (Setaram instruments, SPECANALITICA LDA, Cracavelos, Portugal). Experiments were carried out for spent catalysts using a mass of 20 mg and heating the samples from 20 to 800 °C (heating rate = 10 °C/min) under air flow [47].

### 3.3. Catalytic Runs

#### 3.3.1. Thermal Experiments

CO$_2$ methanation reaction was conducted in a tubular fixed-bed glass U-type reactor (8 mm inner diameter) at atmospheric pressure with a K-type thermocouple inserted in the catalyst bed. Before reaction, catalysts (15mg) were reduced in situ at 500 °C for 1 h with a flow of 5%H$_2$/Ar. Afterwards, a reaction was performed within the temperature range of 250–450 °C, with each temperature being kept for 30 min to reach steady-state operating conditions. The heating ramp between each step was 10 °C/min. An inlet flow of 100 mL/min with the composition of CO$_2$/H$_2$/Ar = 1.5/6/2.5 was used, corresponding

to a GHSV of 52,000 h$^{-1}$. An online micro-chromatograph (Varian GC4900) equipped with a TCD was used to analyze the reaction products. Additionally, stability tests were performed at 250 °C for 5 h, using the same operating conditions of the reaction.

### 3.3.2. Plasma-Assisted Experiments

Plasma catalytic tests were also carried out. Prior to the experiments, catalysts were reduced ex-situ in the presence of 100 mL/min 5%$H_2$/Ar at 500 °C for 1 h. Subsequently, 15 mg of each catalyst was loaded in the DBD plasma reactor, being a cleaning step performed by flowing hydrogen on the surface for 10 min at 12.3 kHz and 21 kV. The same GHSV was kept for the tests, to allow a proper comparison with the results from the thermal experiments. Quartz wool was placed on both sides of the catalyst to fix it, by preventing its mobility. In this study, the feed gas entering the reactor was a $H_2$/$CO_2$ = 4/1 mixture.

In the plasma-catalytic hybrid process, the energy was supplied by a high voltage 30 kHz (MiniPuls 6) and voltages applied ranged from 21 kV to 29 kV (peak to peak). The applied voltage was measured by a digital picoscope (series 3000, PicoTechnology) with a probe (ELDITEST GE 3830). A capacitor (3.32 nF) was inserted between the reactor and the grounded electrode to measure the power provided to the plasma reactor by Q-V Lissajous method [48]. The temperature was measured by a temperature sensor (Pt 100) placed on the outer surface of the quartz tube, in the middle of the ground electrode. The outlet flow was analyzed by a gas chromatograph (Agilent MicroGC 490) equipped with a thermal conductivity detector (TCD). The flowrate was measured with a bubble flowmeter.

For both thermal and plasma conditions, $CO_2$ conversion ($\chi_{CO_2}$) and $CH_4$ selectivity ($S_{CH_4}(\%)$) were calculated by following Equations (1) and (2), respectively.

$$\chi_{CO_2}(\%) = \frac{F_{CO_2\,inlet} - F_{CO_2\,outlet}}{F_{CO_2\,inlet}} \cdot 100 \tag{1}$$

$$S_{CH_4}(\%) = \frac{F_{CH_4\,outlet}}{F_{CO_2\,inlet} - F_{CO_2\,outlet}} \cdot 100 \tag{2}$$

which $F_i$ is the molar flow of each component calculated from the total flowrate and the concentration of gases.

### 4. Conclusions

Ni-based catalysts are proper materials for $CO_2$ methanation. Adding Co as a promoter significantly increased the catalytic performances, such as conversion and methane selectivity both in thermal catalytic system and DBD plasma-catalytic one. The increase in such activity could be explained by the increase in the number of medium basic sites, which are well known to be a key factor for $CO_2$ methanation. The 1Co15Ni/CeO2 was the catalyst which presented the higher number of medium basic sites and showed the best performances both in plasma and non-plasma methanation reaction. A further investigation on electric parameters of the materials will be carried out in order to determine their importance in the plasma assisted methanation reaction. Finally, a deep investigation on Ni dispersion will be carried out by the means of CO chemisorption as reported elsewhere [49] in order to avoid the $H_2$ adsorption on Ceria support.

**Author Contributions:** Conceptualization, C.H. and P.D.C.; methodology, G.H.; M.C.B., C.H. and P.D.C.; validation, M.C.B., C.H. and P.D.C.; formal analysis, G.H.; M.C.B., C.H. and P.D.C.; investigation, G.H.; M.C.B., C.H. and P.D.C.; resources, P.D.C. and C.H.; writing—original draft preparation, G.H.; M.C.B., C.H. and P.D.C.; writing—review and editing, G.H.; M.C.B., C.H. and P.D.C.; visualization, M.C.B.; supervision, M.C.B.; C.H.; P.D.C.; project administration, C.H. and P.D.C.; funding acquisition, P.D.C. and C.H. All authors have read and agreed to the published version of the manuscript.

**Funding:** This research was funded by the European Union's Horizon 2020 research and innovation program, grant number PIONEER-ITN MSCA 813393.

**Acknowledgments:** This work was carried out in the framework of Plasma Catalysis CO₂ Recycling, a PIONEER project which has received funding from the European Union's Horizon 2020 research and innovation programme under the Marie Skłodowska-Curie grant agreement No 813393. Authors thank also to *Fundação para a Ciência e Tecnologia* (FCT) for CQE funding (UIDB/00100/2020 and UIDP/00100/2020) and M. Carmen Bacariza' contract (2020.00030.CEECIND).

**Conflicts of Interest:** The authors declare no conflict of interest.

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
