# Peer review of "On the Effect of Cobalt Promotion over Ni/CeO2 Catalyst for CO2 Thermal and Plasma Assisted Methanation"

_catalysts, doi:10.3390/catal12010036_

Round 1
Reviewer 1 Report
Costa et al. reported thermal CO2 methanation and non-thermal plasma (NTP)-assisted CO2 methanation over 15Ni/CeO2 promoted with 1 and 5 wt% of cobalt. The promotion effect of cobalt is proven both for plasma and thermal reaction and can be linked mostly with the basic properties of the materials. The physiochemical properties of these catalysts were characterized by XRD, H2-TPR, TEM, etc. Some interesting observations and scientific viewpoints are included in the manuscript. However, there are still some questions need to be further addressed before it can be accepted for publication. Some comments are given as follows:
(1) The element contents of these catalysts should be supplemented by ICP-OES.
(2) The TEM qualities of these catalysts is very bad and they should be re-done.
(3) The Ni dispersion should be supplemented by chemisorption such as the use of H2 or CO to further compare the intrinsic activities of these catalysts.
(4) The Ni and Co value states of these catalysts should be supplemented by XPS.
Author Response
(1) The element contents of these catalysts should be supplemented by ICP-OES.
We agree with the reviewer, however, by lack of time and due to pandemic situation, it is impossible for us to do these types of experiments. We would like to point out that in a previous study using the same loading of Ni, 14.8wt.% was found for a theoretical value of 15.
(2) The TEM qualities of these catalysts is very bad and they should be re-done.
We thank you for your comment, the problem is not the TEM analysis itself it is the fact that the format had to be changed to introduce it in the template. Enclosed here as example pictures with better resolution from 15Ni/CeO2 catalyst and 1Co15Ni/CeO2 catalyst.
(3) The Ni dispersion should be supplemented by chemisorption such as the use of H2 or CO to further compare the intrinsic activities of these catalysts.
We agree with the reviewer; however, we do not have the possibility to do it. These techniques proposed by the reviewer will be done within the mechanistic study that we are currently investigating.
(4) The Ni and Co value states of these catalysts should be supplemented by XPS.
We agree and this will be done prior after mechanistic study in presence and absence of plasma. Thus, this will be the purpose of the next article on Co/Ni Ceria based catalysts for methanation.

Reviewer 2 Report
The paper “On the effect of Cobalt promotion over Ni/CeO2 catalyst for 2 CO2 thermal and plasma assisted methanation” by Golshid Hasrack, M. Carmen Bacariza, Carlos Henriques, Patrick Da Costa, present the information about CO2 hydrogenation to synthetic fuels and value-added molecules as a promising technology to stabilize anthropogenic greenhouse gas emissions, being performed over 15Ni/CeO2 promoted with 1 and 5 wt% of cobalt.
The paper can be published after minor correction. However, the authors can take into account the following:
- please be careful with the correct numbering of the bibliography
- it would be advisable to replace "basic sites" with "base sites"
- it is at least surprising to me that CeO2 does not have strong basicity values while Ni, which does not have a low electronegativity value comparing to Ce, has strong basicity
- please change “diffraction peaks” with “diffraction lines” or another
- image quality needs to be improved
Author Response
(1) Please be careful with the correct numbering of the bibliography
We carefully checked the bibliography, thanks to the reviewer.
(2) It would be advisable to replace "basic sites" with "base sites"
In the literature, enclosed examples below (a,b,c), the sites counted by CO2 are basic ones. In the literature, all the authors always consider basic sites, we then decided to keep that.
(a)Summa, P.; Åšwirk, K.; Wang, Y.; Samojeden, B.; Rønning, M.; Hu, C.; Motak, M.; Da Costa, P. Effect of Cobalt Promotion on Hydrotalcite-Derived Nickel Catalyst for CO2 Methanation. Appl. Mater. Today 2021, 25, 101211, doi:10.1016/j.apmt.2021.101211.
(b)Liu, Q.; Bian, B.; Fan, J.; Yang, J. Cobalt Doped Ni Based Ordered Mesoporous Catalysts for CO2 Methanation with Enhanced Catalytic Performance. Int. J. Hydrog. Energy 2018, 43, 4893–4901, doi:10.1016/j.ijhydene.2018.01.132.
(c)Ge, Y.; He, T.; Han, D.; Li, G.; Zhao, R.; Wu, J. Plasma-Assisted CO2 Methanation: Effects on the Low-Temperature Activity of an Ni–Ce Catalyst and Reaction Performance. R. Soc. Open Sci. 6, 190750, doi:10.1098/rsos.190750.
(3) It is at least surprising to me that CeO2 does not have strong basicity values while Ni, which does not have a low electronegativity value comparing to Ce, has strong basicity
It is common to find on Ceria or Ceria Zirconia based materials low basicity. As already shown by Mikhail et al. on Ceria-Zirconia based material or also by Nizio et al. on the similar materials
(d)Nizio, M.; Albarazi, A.; Cavadias, S.; Amouroux, J.; Galvez, M.E.; Da Costa, P. Hybrid Plasma-Catalytic Methanation of CO2 at Low Temperature over Ceria Zirconia Supported Ni Catalysts. Int. J. Hydrog. Energy 2016, 41, 11584–11592, doi:10.1016/j.ijhydene.2016.02.020.
(e)Nizio, M.; Benrabbah, R.; Krzak, M.; Debek, R.; Motak, M.; Cavadias, S.; Gálvez, M.E.; Da Costa, P. Low Temperature Hybrid Plasma-Catalytic Methanation over Ni-Ce-Zr Hydrotalcite-Derived Catalysts. Catal. Commun. 2016, 83, 14–17, doi:10.1016/j.catcom.2016.04.023.
(f)Mikhail, M.; Da Costa, P.; Amouroux, J.; Cavadias, S.; Tatoulian, M.; Gálvez, M.E.; Ognier, S. Tailoring Physicochemical and Electrical Properties of Ni/CeZrOx Doped Catalysts for High Efficiency of Plasma Catalytic CO2 Methanation. Appl. Catal. B Environ. 2021, 294, 120233, doi:10.1016/j.apcatb.2021.120233.
Also According to the literature (g,h) addition Ni into the ceria or ceria-zirconia improved the redox properties of the material favoring oxygen transfer, which would be required for CO2 activation [26], [27].
(g) J. Gao, J. Guo, D. Liang, Z. Hou, J. Feia, X. Zheng, Production of syngas via autothermal reforming of methane in a fluidized-bed reactor over the combined CeO2–ZrO2/SiO2 supported Ni catalysts, Int J Hydrogen Energy, 33 (2008), p. 5493
- Radlik, M. Adamowska-Teyssier, A. Krztoń, K. Kozieł, W. Krajewski, W. Turek, et al.
(h)Dry reforming of methane over Ni/CeZrO2 catalysts: effect of Ni loading on the catalytic activity and H2/CO production, C R Chim (2015), Volume 18, Issue 11, November 2015, Pages 1242-1249
Also, it was suggested by summa et al. (i) that regarding the addition of Ni to the support, the metallic nickel 3d orbits are not filled, which results in activate CO2 to form CO2−. Nickel form a complex with the hydroxyl group on the surface of the carrier, as a Lewis base site, reacting with CO2 and shifting the desorption temperature to a higher range and increasing the base sites. The same behavior is observed with addition of Co.
(i) Summa, P.; Åšwirk, K.; Wang, Y.; Samojeden, B.; Rønning, M.; Hu, C.; Motak, M.; Da Costa, P. Effect of Cobalt Promotion on Hydrotalcite-Derived Nickel Catalyst for CO2 Methanation. Appl. Mater. Today 2021, 25, 101211, doi:10.1016/j.apmt.2021.101211.
(4) Please change “diffraction peaks” with “diffraction lines” or another
We agree with the remark of the reviewer and we have changed it. Thanks!
(5) Image quality needs to be improved
We try to improve quality in changing the format of the pictures, we ll change to PNG format.

Round 2
Reviewer 1 Report
Some data did not be supplemented in the revised manuscript, but considering the COVID-19's pandemic, the article can be accepted after citing the reference (please the authors read the charateriaztion techniques in this article: EfficientHydrogenation of Xylose and Hemicellulosic Hydrolysate to Xylitol over Ni-Re Bimetallic Nanoparticle Catalyst, Nanomaterials, 2020, 10, 73. https://doi.org/10.3390/nano10010073 )
Author Response
Dear Reviewer,
As suggested, we have decided to perform in the next month CO chemisorption in order to confirm the potential link between Ni particle size/dispersion and the catalytic activity in absence and presence of Plasma.
The reference is now 49.
Reference 49 : Efficient Hydrogenation of Xylose and Hemicellulosic Hydrolysate to Xylitol over Ni-Re Bimetallic Nanoparticle Catalyst, Nanomaterials, 2020, 10, 73. https://doi.org/10.3390/nano10010073 )